# Lyophilized powder of calf bone marrow hydrolysate liposomes improved renal anemia: *In vitro* and *in vivo* evaluation

**Li Li, Shasha Zhao, Xiaodun Liu, Zhe Xu, Dong Li, Xiaoyu Dai**📷*

Department of Research and Development, Jinan Perfect Biological Technology Co., LTD, Jinan, Shandong, China

* daixiaoyu2024@163.com

**Data Availability Statement:** All relevant data are within the manuscript and its Supporting information files.

## Abstract

This study aimed to find whether oral administration of calf bone marrow hydrolysate liposomes (CBMHL) can improve renal anemia. Calf bone marrow was defatted, papain hydrolyzed, liposomalized and lyophilized. Its hematopoietic ability was proved by the colony formation experiment of umbilical cord blood hematopoietic stem cells *in vitro*. The rat model of renal anemia was established by adenine intragastric administration, and different concentrations of CBMHL were intragastricly administrated. Blood routine and serological indexes, transcription levels of hematopoietic factors and renal pathology were detected. From the appearance, redispersability, water content, liposome indexes and stability of Lyophilized powder of CBMHL, it could be concluded that the quality of freeze-dried CBMHL powder under this freeze-drying process was good. Compared with the control group, the burst forming unit-erythroid (BFU-E) in the CBMHL group was larger and the number of colonies increased significantly in the colony formation experiment ($P <$ 0.05). The results of lyophilized powder of CBMHL co-culture with human adipose mesenchymal stem cells (MSCs) and human cytokine-induced killer (CIK) cells showed that the lyophilized powder of CBMHL had no potential toxicity and allergic reaction *in vitro*. Compared with the Model Group, the red blood cell (RBC) count, hemoglobin (HB) content and hematokrit (HCT) of rats blood routine in the Model+high doses of CBMHL Group (Model +H-CBMHL Group) increased significantly ($P <$ 0.05). Serum erythropoietin (EPO) and glutathione (GSH) levels increased significantly ($P <$ 0.05), while serum creatinine (Cr) levels decreased significantly ($P <$ 0.05). The transcription level of *Epo* in kidney increased significantly ($P <$ 0.05), the transcription levels of erythropoietin receptor (*Epor*) in bone marrow and interleukin 6 (*Il6*) in spleen were significantly increased ($P <$ 0.01). The fragility of red blood cells decreased significantly, and the pathological structure of kidney improved significantly. It was proved that lyophilized powder of CBMHL could effectively enhance the hematopoietic ability of rats with renal anemia and protect the kidney structure and function.

**Funding:** This work was supported by Shandong Province Key R&D Program (Medical Food Special Program) Project (No.2019YYSP009).

## 1. Introduction

Renal anemia is a common complication of chronic kidney disease, with a rate of over 98% in end-stage cases. It can cause hypoxia, fatigue, cognitive issues, malnutrition, and heart problems, and worsen the course of the kidney disease. The progression of kidney disease significantly reduces both the survival and quality of life of affected patients [1, 2]. Current medical research suggests that the main causes of this disease are reduced production of erythropoietin (EPO), iron deficiency, and metabolic imbalances. Other contributing factors include malnutrition, oxidative stress, inflammation, and increased fragility of red blood cells. While traditional Chinese medicine does not formally recognize renal anemia, it believes that deficiency in the spleen and kidney is at the core of the disease [3, 4].

Currently, the clinical treatment of renal anemia often involves the injection of recombinant human erythropoietin, which is only effective for some patients. Long-term use can increase blood pressure and induce various cerebrovascular diseases. As a new type of hypoxia-inducible factor prolyl hydroxylase inhibitor, Roxadustat has good clinical effects in improving renal anemia with minimal side effects. However, its high price prevents widespread use. Levocarnitine, iron preparations, and other drugs are only effective when used in combination with EPO. Traditional Chinese medicine, such as Prescription Yishen Qingli Huoxue, has achieved good clinical results in the treatment of renal anemia, but it takes time to take effect and has a poor taste [4–9]. Therefore, it is very important to find a product with minimal side effects, fast efficacy, affordable price, and the ability to promote hematopoiesis fundamentally.

Some studies have shown that bone marrow contains phosphoproteins, chondroitin sulfate, growth factors, hematopoietic factors, small molecule nucleic acids, various amino acids, multiple vitamins, and trace elements (iron, zinc, copper, strontium, etc.), making it a valuable natural medicinal and food resource [10, 11]. In traditional Chinese medicine, cow bone marrow is believed to have the functions of moistening the lungs, strengthening muscles and bones, nourishing the kidneys, and filling the marrow. The medical book "Lvshantang Debate" states: "The kidneys are the water organ, responsible for storing essence and transforming blood." This means that if the kidney essence is sufficient, the blood will be abundant. If the kidney essence is insufficient, the production of essence and blood will decrease, leading to anemia. Therefore, the key to treating anemia is to nourish the kidneys. From this, it can be inferred that cow bone marrow may be a potential blood-nourishing resource. As we all know, with age, the amount of red bone marrow in the bone marrow cavity gradually decreases, while the amount of yellow bone marrow increases. Therefore, newborn cow bone marrow contains more abundant hematopoietic substances than ordinary bone marrow [12]. At the same time, as an important source of active peptides, the enzymatic hydrolysate not only has simple nutritional functions but also has certain physiological regulatory functions [13]. Studies have shown that peptide forms of amino acids are more easily and rapidly absorbed by the intestine than free amino acids, increasing the utilization rate of amino acids by the body [14]. Liposomes are a new type of carrier that wraps biologically active substances such as vitamins, minerals, peptides, and enzymes with membrane materials such as cholesterol and phospholipids. They reduce the influence of gastric and intestinal acidic media and enzymes, and the lipid bilayer structure is similar to biological membranes, enhancing the endocytosis of intestinal epithelial cells, making it easier for active substances to pass through the intestinal epithelium and enter the systemic circulation, thereby improving their bioavailability [15]. In addition, liquid liposomes have a short storage time, are prone to deterioration, and are inconvenient to transport. The use of freeze-drying to produce solid powder enhances the stability of

liposomes, making them easier to store and transport, while not compromising the biological activity of hematopoietic substances [16].

In summary, this study planed to first select newborn calf bone marrow with strong hematopoietic function as the raw material, performed defatting and hydrolysis, encapsulated it with liposome technology, and then used vacuum low-temperature freeze-drying technology to prepare calf bone marrow hydrolysate liposomes (CBMHL) freeze-dried powder. Then, the colony culture experiment was used to test its in vitro hematopoietic support ability, and finally, the rat adenine-induced renal anemia model was used to verify its auxiliary hematopoietic function in vivo through oral administration.

## 2. Materials and methods

### 2.1. Materials and instruments

Calf bone marrow comed from Daqing Chunqiu Biotechnology Co., Ltd.; Umbilical cord blood was normal pregnancy umbilical cord blood from the Department of Obstetrics and Gynecology of Qilu Hospital, Shandong University, China; All parties were informed and this study have obtained the approval of the Research Ethics Committee of Qilu Hospital, Shandong University, China (Approval No. KYLL-2020(KS)-4018, S1 File); Wistar rats, female, 36 in total, weighing 180–200 g, SPF grade, were purchased from Sbeifu (Beijing) Biotechnology Co., Ltd., tested and qualified by Sbeifu (Beijing) Biotechnology Co., Ltd. on July 9, 2020, animal license number: SCXK (京) 2019–0010, quality license number: 1102290038808 (S1 Fig). This study had been reviewed by the Laboratory Animal Welfare Ethics Committee of Qilu Hospital, Shandong University, China (Approval No. DWLL-2020-2535, S2 File); Lecithin (soybean), cholesterol, and adenine were from Solarbio Biotechnology Co., Ltd.; Quan'an Su milk powder was from Abbott Pharmaceutical Co., Ltd.; Recombinant Human EPO was from PeproTech, USA; Semi-solid H4434 medium was from STEMCELL Technologies, Canada; Rat Cr ELISA KIT, Rat BUN ELISA KIT were from Shanghai Enzyme-linked Biotechnology Co., Ltd.; Reduced glutathione (GSH) detection kit was from Yacoo Biotechnology Co., Ltd.; Rat EPO ELISA Research Reagent was from BOSTER, USA; Red blood cell osmotic fragility test kit (sieve method) was from Shanghai Yuan Ye Biotechnology Co., Ltd.; Improved HE (hematoxylin-eosin) staining kit was from Solarbio; Reverse transcription kit FSQ-101, SYBR Green realtime QPK-201 were from TOYOBO, Japan.

N-1300 rotary evaporator was from EYELA, Japan; JY92-IIDN ultrasonic cell crusher was from Shanghai Huxi Industrial Co., Ltd.; NanoZS nanoparticle potential analyzer was from Malvern, UK; LGJ-30FG vacuum freeze dryer was produced by Cangzhou Borui Kangheng Electromechanical Equipment Co., Ltd.; HF240 incubator was produced by Shanghai Likang Group; CKX53 microscope was from Olympus, Japan; BM21B semi-automatic blood cell analyzer was from Beijing Baolingman Sunshine Technology Co., Ltd.; MULTISKAN MK3 microplate reader was from Thermo Fisher Scientific, USA; UER 2005012 UV-visible spectrophotometer was from Shanghai MAPADA; SMA4000 DNA/Protein Analyzer was from Beijing Merinton; qTOWER3G fluorescence quantitative gene amplification instrument was from Jena, Germany.

### 2.2. Experimental methods

**2.2.1. Enzymatic hydrolysis process of calf bone marrow.** Hematopoietic active substances: Calf bone marrow was subjected to defatting treatment using low temperature density defatting method, and Soxhlet extraction method was used to determine the defatting rate of bone marrow [17]. After defatting, the bone marrow was added with papain enzyme at a ratio of 5 mg/g, pH 6.0, and enzymatically hydrolyzed at 37 °C for 8 hours [18]. The enzymatic

hydrolysate was filtered through a 100 μm mesh to remove impurities. Formaldehyde titration method was used to determine the hydrolysis rate and the content of amino nitrogen. Then, the hydrolysis rate was calculated using the following formula:

$$DH(\%) = \frac{AN}{TN} \times 100$$

DH represents the protein hydrolysis rate, %; AN represents the content of amino acids in the hydrolysate, %; TN represents the total nitrogen content in the unhydrolyzed protein, %.

**2.2.2. Preparation of liposomes.** The classic thin film hydration technique was adopted for the preparation of liposomes [19]. Soybean lecithin and cholesterol were mixed in a 3:1 mass ratio and dissolved in a fixed amount of chloroform. The mixture was placed on a rotary evaporator at 45°C in a constant temperature water bath, with a rotation speed of 100 r/min for 10 minutes. The solvent was evaporated under reduced pressure to form a lipid film. The calf bone marrow hydrolysate-PBS buffer was heated in a constant temperature water bath at 45 °C and was poured into the rotary evaporator containing the lipid film (calf bone marrow hydrolysate: soybean lecithin, 1:15 mass ratio). The mixture was hydrated at 45 °C with a rotation speed of 100 r/min for 30 minutes. The liquid was transferred to a beaker and sonicated using an ultrasonic cell disruptor for 8 minutes (200 W, 1s on, 1s off) to obtain liposome samples. We characterized CBMHL in terms of appearance, particle size, dispersion, encapsulation efficiency (EE%) and morphology.

**2.2.3. Liposomes freeze-drying.** The liposomes were freeze-dried using the classic freeze-drying process [20, 21]. Added 5 g of mannitol to every 100 mL of liposome sample. Freeze-dry according to the following freeze-drying curve (Fig 1): decreased from room temperature to -15 °C at a rate of 2 °C/min, stabilized for 80 min, then decreased to -30 °C at a rate of 2 °C/min and maintained for 90 min, then decreased to -40 °C at a rate of 2 °C/min and maintained for 20 min, then increased to -5 °C at a rate of 2 °C/min, evacuate to 25 Pa, lyophilized for 6 h, then increased to 10 °C at a rate of 2 °C/min, evacuated to 10 Pa, maintained for 8.5 h, then increased to 20 °C at a rate of 1 °C/min, evacuated to 3 Pa, maintained for 3.5 h.

**2.2.4. Stability study of liposomes after freeze-drying.** The liquid liposomes and freeze-dried liposomes were placed in a cool cabinet at 4 °C, and taken out after 8 weeks. 2ml of pure water was added to freeze-dried powder to redissolve. The particle size, dispersity, and EE% were measured respectively to investigate the stability of lyophilized liposomes.

**2.2.5. The effect of lyophilized powder of CBMHL on the apoptosis of human adipose mesenchymal stem cells (MSCs) and the proliferation of human cytokine-induced killer (CIK) cells.** The MSCs of human adipose tissue were isolated and cultured using collagenase digestion [22], and passaged at a ratio of 1:3. After the fourth generation of passage, the MSCs culture with a plating efficiency of about 85% were incubated with different concentrations of 0 mg/L, 0.05 mg/L, 0.5 mg/L, and 5 mg/L of CBMHL at 37 °C and 5% CO2. After 48 hours of cultivation, they were harvested and subjected to apoptosis detection using the ANNEXIN V-FITC/PI cell apoptosis detection kit according to the instructions.

According to the method in the reference, mononuclear cells (MNCs) from human peripheral blood were isolated and induced to CIK [23]. After 12 days, CIK cells were harvested and four culture media with different concentrations of CBMHL (the final mass concentrations of CBMHL were 0 mg/L, 0.05 mg/L, 0.5 mg/L, and 5 mg/L) were prepared. The CIK cells were introduced to these four concentrations of culture media, and the final cell density was $5\times10^5$/mL. The cell solution were inoculated into a 96-well plate at a volume of 100 μL/well, and incubated for 12, 24, 36, and 48 hours at 37 °C and 5% CO2. The absorbance value at 450 nm was measured according to the instructions of the CCK-8 kit.

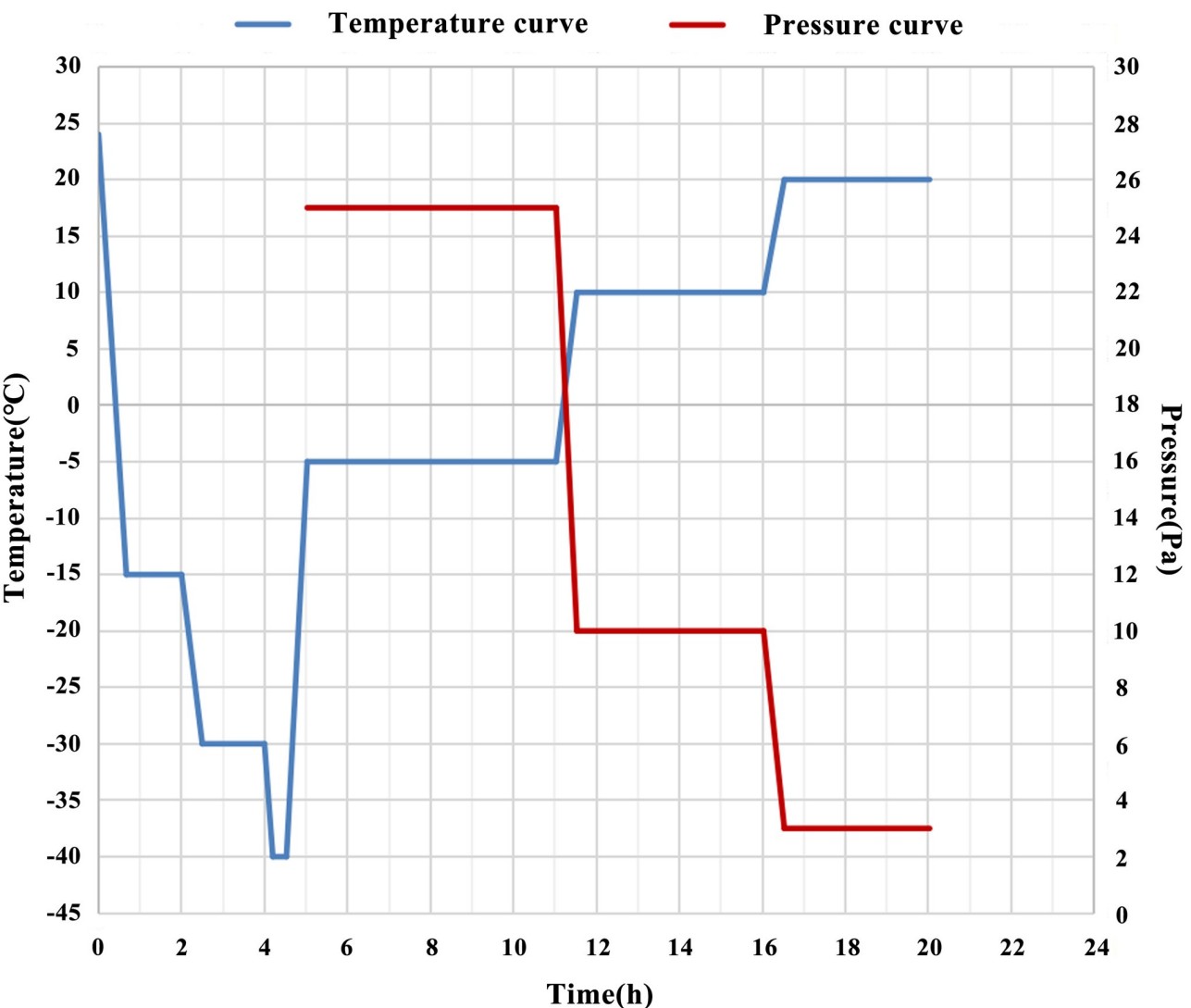

**Fig 1. The freeze-drying curve of liposomes.**

**2.2.6. Hematopoietic stem cell colony formation experiment.** Weighed 2 g of, added 20 mL of deionized water and stirred to dissolve. Filtered through coarse filter paper and then a 0.22 μm filter, divided into aliquots and stored frozen. MNCs were separated from human anticoagulant cord blood using gradient density centrifugation [24]. The cells were resuspended in PBS buffer to make a cell suspension with a density of $1.1 \times 10^5$/mL. For each 5 mL of semi-solid culture medium, added 0.25 mL of cell suspension and 0.25 mL of filtered lyophilized powder of CBMHL solution at different concentrations (diluted 10, 100, and 1000 times). Mixed well. These were designated as CBMHL samples 1, 2, and 3 groups (with decreasing concentrations of filtered solution), and PBS buffer as a blank control. The four groups were seeded in a 6-well plate, with 1.1 mL per well, and incubated at 37 ˚C with 5% $CO_2$ for 14 days. After incubation, taked photos and counted the cells.

**2.2.7. Establishment and grouping of renal anemia rat model.** Rats were housed in a SPF environment with a temperature of 24 ± 1˚C, humidity of 50 ± 10%, and a 12-hour light-

dark cycle. After 4 days of adaptation, the rats were randomly divided into a normal control group (6 rats) and a modeling group (30 rats). The modeling group was orally administered a 2% adenine water solution at a dose of 250 mg/kg/d [25, 26], while the normal control group was given the same volume of deionized water once a day for 28 days. All rats were anesthetized by inhaling isoflurane in a gas anesthesia machine, and blood samples were taken from the inner canthus for blood routine analysis: red blood cell (RBC) count, hemoglobin (HB) content, and hematocrit (HCT). Serum was obtained from centrifugation of peripheral blood at 3000 r/min for 30 min for renal function tests: serum creatinine (Cr) and blood urea nitrogen (BUN). The results confirmed the characteristics of renal anemia, indicating successful modeling. The renal anemia rats were randomly divided into 5 groups, with 6 rats in each group.

All the rats in the experiment were grouped as follows: Normal Group, Model Group, Model+EPO Group, CBMHL oral groups (Model+low doses of CBMHL Group (Model+-L-CBMHL Group), Model+medium doses of CBMHL Group (Model+M-CBMHL Group), Model+high doses of CBMHL Group (Model+H-CBMHL Group)). After grouping, each group was given a dose of 1 mL/100 g BW for 21 consecutive days. After successful modeling, the physical condition of rats in the Model Group was weak. In order to prevent rats' dying that would make the experiment unable to proceed, the mortality rate of the rats was reduced by feeding them with nutritionally rich Quan'an Su milk powder through gavage. Both the Normal Group and the Model Group were only fed with Quan'an milk powder at a dose of 0.28 g/kg/d. In the Model+EPO Group, in addition to feeding with Quan'an milk powder, EPO was also injected subcutaneously at a dose of 250 IU/kg [27], once every 3 days for a total of 7 injections. In the Model+L-CBMHL Group, in addition to feeding with Quan'an milk powder, lyophilized powder of CBMHL was also administered by gavage at a dose of 0.14 g/kg/d. In the Model+M-CBMHL Group, in addition to feeding with Quan'an milk powder, lyophilized powder of CBMHL was also administered by gavage at a dose of 0.28 g/kg/d. In the Model+H-CBMHL Group, in addition to feeding with Quan'an milk powder, lyophilized powder of CBMHL was also administered by gavage at a dose of 0.56 g/kg/d.

**2.2.8. Sample collection.**   After the administration, the rats in each group were anesthetized by inhaling isoflurane in a gas anesthesia machinethe. After anesthesia, blood was taken from the inner canthus and 0.5 mL was collected in a 1.5 mL EP tube containing anticoagulant. Red blood cell fragility test and blood routine analysis were performed according to the instructions of the red blood cell osmotic fragility test kit. Another 1.5 mL of whole blood was collected, placed at 4°C overnight, centrifuged at 3500 rpm for 10 min, and the serum was collected and divided, and stored at -80 °C for later use. Serum EPO and GSH were detected according to the instructions of the test kit. After dislocation of the cervical vertebra, routine dissection was performed quickly, one side of the kidney was taken, fixed with 4% paraformaldehyde, washed, dehydrated, cleared, embedded, and sectioned with a thickness of 4 μm. Routine rehydration, HE staining, and photography were performed. The other side of the kidney and spleen were both added to RNAlater and stored in a -80 °C freezer. The femurs were cut at both ends and flushed with physiological saline to extract bone marrow. After homogenization, the marrow was placed on the surface of lymphocyte separation solution with a density of 1.084, centrifuged at 2400 rpm for 25 min, and the white film layer was taken. It was centrifuged with physiological saline at 1500 rpm for 10 min twice, and the precipitate was added to RNAlater and stored in a -80 °C freezer.

**2.2.9. Quantitative PCR detection of the expression of hematopoietic factors and their receptors.**   The single nuclear cells from the kidney, spleen, and bone marrow were taken, added Trizol and RNA was extracted using a standard procedure. After reversed transcription into cDNA, the reaction system was set to 20 μL per well according to the instructions of the

**Table 1. Primer sequences for quantitative RT-PCR.**

| Genes | ID | Sequence(5'→3') | | Tm(˚C) | Product length(bp) |
|---|---|---|---|---|---|
| *Gapdh* | NM_017008.4 | forward | TTGTGCAGTGCCAGCCTC | 60.6 | 193 |
| | | reverse | AACTTGCCGTGGGTAGAGTC | 59.68 | |
| *Epor* | NM_017002.2 | forward | GGACCCTCTCATCTTGACGC | 60.18 | 185 |
| | | reverse | GCAACAGCCATAGCTGGAAGT | 60.95 | |
| *Epo* | NM_017001.1 | forward | GGGGGTGCCCGAACG | 59.65 | 124 |
| | | reverse | TACCTCTCCAGAACGCGACT | 60.32 | |
| *Il6* | NM_012589.2 | forward | CATTCTGTCTCGAGCCCACC | 60.46 | 91 |
| | | reverse | GCTGGAAGTCTCTTGCGGAG | 60.74 | |

SYBR Green PCR kit. The transcription levels of kidney erythropoietin (*Epo*), spleen interleukin 6 (*Il6*), and bone marrow erythropoietin receptor (*Epor*) were detected on the qTOWER3G fluorescence quantitative gene amplifier from Jena, Germany. The reference gene was glyceraldehyde-3-phosphate dehydrogenase (*Gapdh*). The reaction conditions were 95 ˚C for 1 min, 95 ˚C for 15 s, 65 ˚C for 45 s, and 40 cycles. The relative expression of the target gene was represented by $2^{-\Delta\Delta Ct}$ ($\Delta\Delta Ct$ = (Ct target gene—Ct reference gene) experimental group—(Ct target gene—Ct reference gene) control group). The relative expression level of the gene in the normal group was set to 1. The primer sequences were shown in Table 1.

### 2.3. Data analysis

All data were presented as mean ± standard deviation (± s) and analyzed using SPSS 26.0 software for statistical analysis. Independent sample t-test was used to compare the data between the two groups. For multiple group comparisons, one-way analysis of variance (ANOVA) was first used to detect overall differences among the groups. If significant differences were found, the LSD method was used for pairwise statistical analysis. $P < 0.05$ was considered statistically significant, and $P < 0.01$ was considered highly significant with statistical significance.

## 3. Results

### 3.1. Liposome characterization and stability

The average defatting rate of the low-temperature density defatting method detected by Soxhlet extraction method was 99.5%. The average protein hydrolysis rate obtained by enzymatic hydrolysis at an enzyme dosage of 5 mg/g, pH 6.0, and 37 ˚C for 8 h was 75.70%. According to the naked eye observation, the synthesized liposomes were white, bluish and translucent emulsion (Fig 2A); The average particle size of liposomes was (46.98 ± 0.67) nm and the PDI was 0.235 ± 0.011 detected by nanoparticle size analyzer (Fig 2C and Table 2); The concentration of polypeptide inside and outside the liposomes was detected by UV spectrophotometry, and the encapsulation efficiency was calculated, the EE% was (93.13 ± 0.38)% (Table 2); TEM was used to observe the morphology of liposomes, which were uniform spherical or quasi spherical, and were a kind of micro vesicles with lipid bilayer (Fig 2B).

After lyophilized, as shown in S2 Fig, the lyophilized powder of liposomes had a plump and uniform appearance, and there was no color difference from top to bottom; The powder dissolved rapidly, the rehydration time was less than 2 s, and the liquid color was homogenous; The water content of liposomes after lyophilized was < 1.8%.

After storage at 4 ˚C for 8 weeks, as shown in Table 2, the average particle size and PDI of liquid liposomes increased significantly (P < 0.01), the EE% decreased significantly (*P* < 0.01),

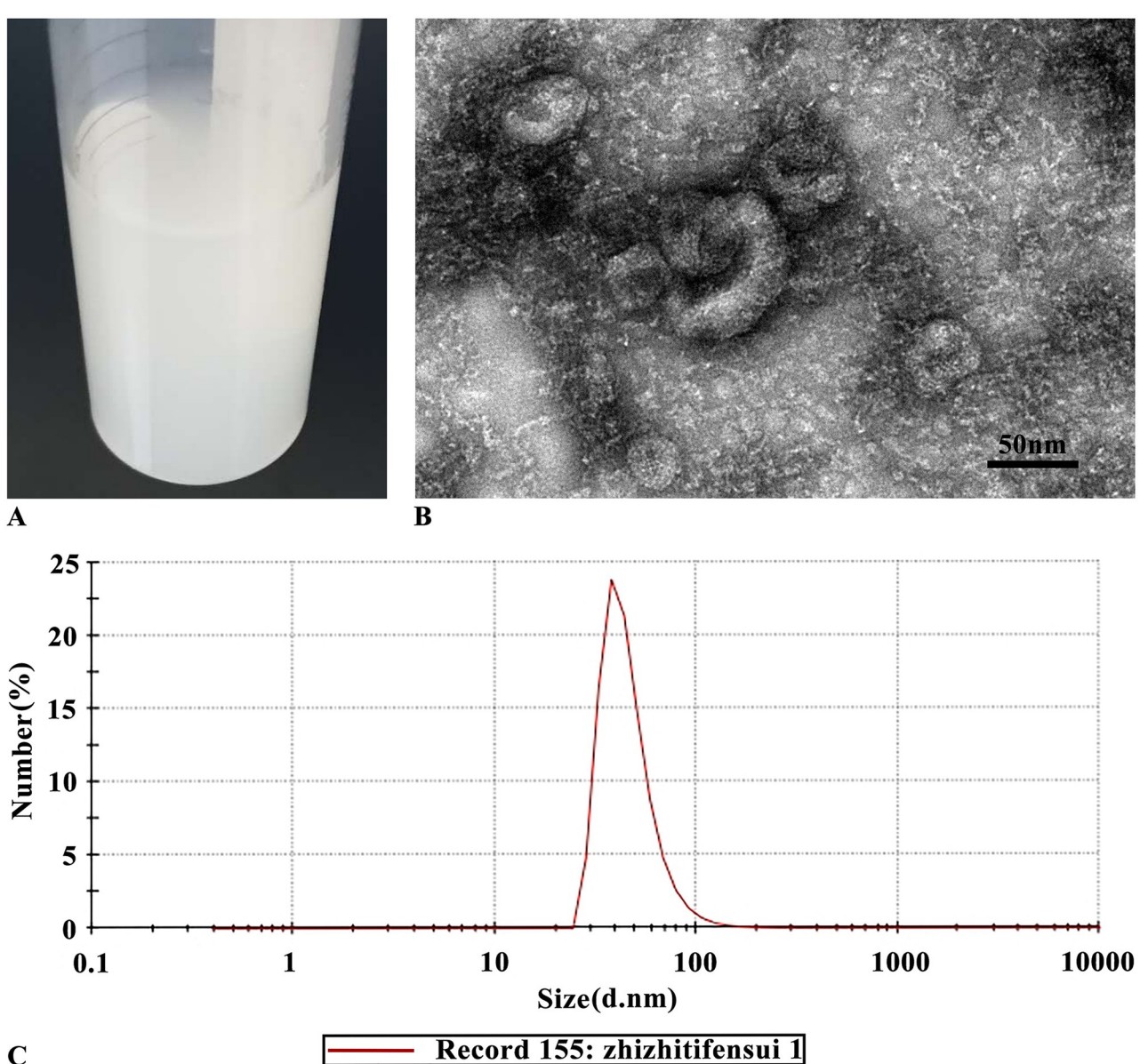

**Fig 2. The liposomes characterization.** (A) appearance. (B) Tem image. (C) Particle size distribution.

**Table 2. The particle size, PDI and EE% of the different liposome samples.**

| Name of the sample | Particle size(nm) | PDI | EE% |
|---|---|---|---|
| Initial liposomes | 46.98±0.67 | 0.235±0.011 | 93.13±0.38 |
| Liposomes after 8 weeks | 68.83±1.14** | 0.398±0.018** | 86.65±0.92** |
| liposome lyophilized powder after 8 weeks | 47.12±0.22 | 0.206±0.007 | 92.26±0.58 |

Data expressed as mean ± SD, n = 3.

**$P < 0.01$ *vs*. Initial liposomes.

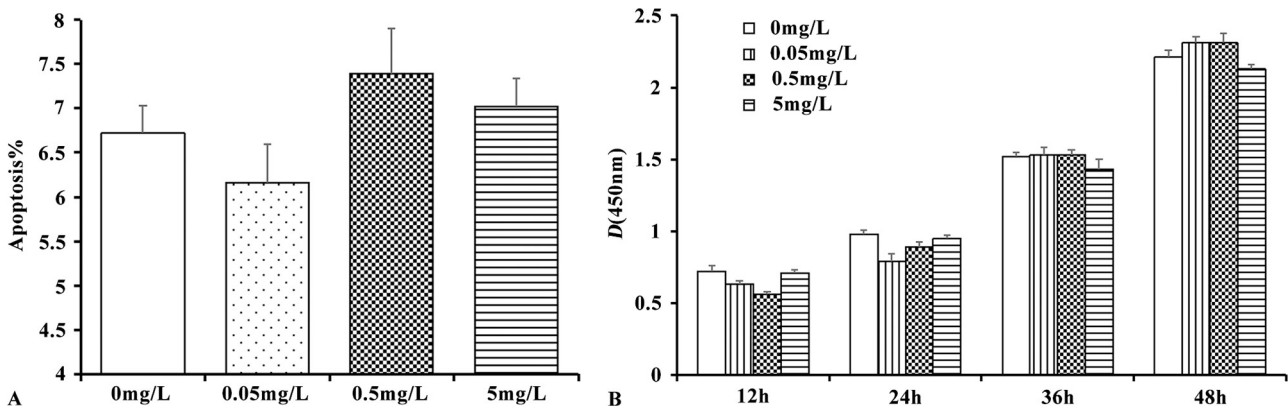

**Fig 3. In vitro toxicity study of freeze-dried powder of CBMHL.** (A) The effect of CBMHL on the apoptosis of human adipose MSC. (B) The effect of CBMHL on the proliferation of human CIK cells. (Data expressed as mean ± SD, $n$ = 3).

and the average particle size, PDI and EE% of freeze-dried liposomes had no significant changes, indicating that the stability of liposomes was significantly enhanced after freeze-drying.

### 3.2. Study on the in vitro toxicity of lyophilized powder of CBMHL

Human MSCs were co-cultured with lyophilized powder of CBMHL for 48 hours. The apoptosis rate of MSCs in each group was detected using an apoptosis detection kit. The results (Fig 3A) showed that there was no significant difference in the apoptosis rate of MSCs among the four groups of 0 mg/mL, 0.05 mg/mL, 0.5 mg/mL, and 5 mg/mL ($P > 0.05$), indicating that the freeze-dried powder of CBMHL did not inhibit the apoptosis of human adipose MSCs in vitro; Different concentrations of lyophilized powder of CBMHL were added to the human CIK culture, and the proliferation of CIK cells in each group was detected using the CCK-8 kit after co-culturing for 12 h, 24 h, 36 h, and 48 h. The results were shown in the Fig 3B. There was no significant difference in the proliferation of CIK cells in each co-culturing group during different period ($P > 0.05$), indicating that the freeze-dried powder of CBMHL had no effect on the proliferation ability of human immune cells in vitro, neither promoting nor inhibiting.

### 3.3. Hematopoietic colony growth statistics

When hematopoietic colonies were cultured for 14 days, the morphology of the colonies in each group could be seen in Fig 4. The majority of the colonies in each group were burst forming unit-erythroid (BFU-E) colonies, which contained hemoglobin and appear red in color. The colony volume increased significantly in CBMHL samples 1, 2, and 3 compared to the control group, with CBMHL sample 1 having the largest colony volume and the highest number of red blood cells. The statistical analysis of colony numbers in each group was shown in Table 3. Compared to the control group, the number of BFU-E colonies increased significantly in CBMHL sample 1 group ($P < 0.05$), indicating that lyophilized powder of CBMHL had a hematopoietic-promoting function, especially in erythroid hematopoiesis.

### 3.4. lyophilized powder of CBMHL could improve the indicators of the rat model of renal anemia

After modeled, compared with the Normal Group, the rats in the Model Group showed rough and dull hair, hair loss, emaciation, hunchback, huddling, lethargy, increased urine output,

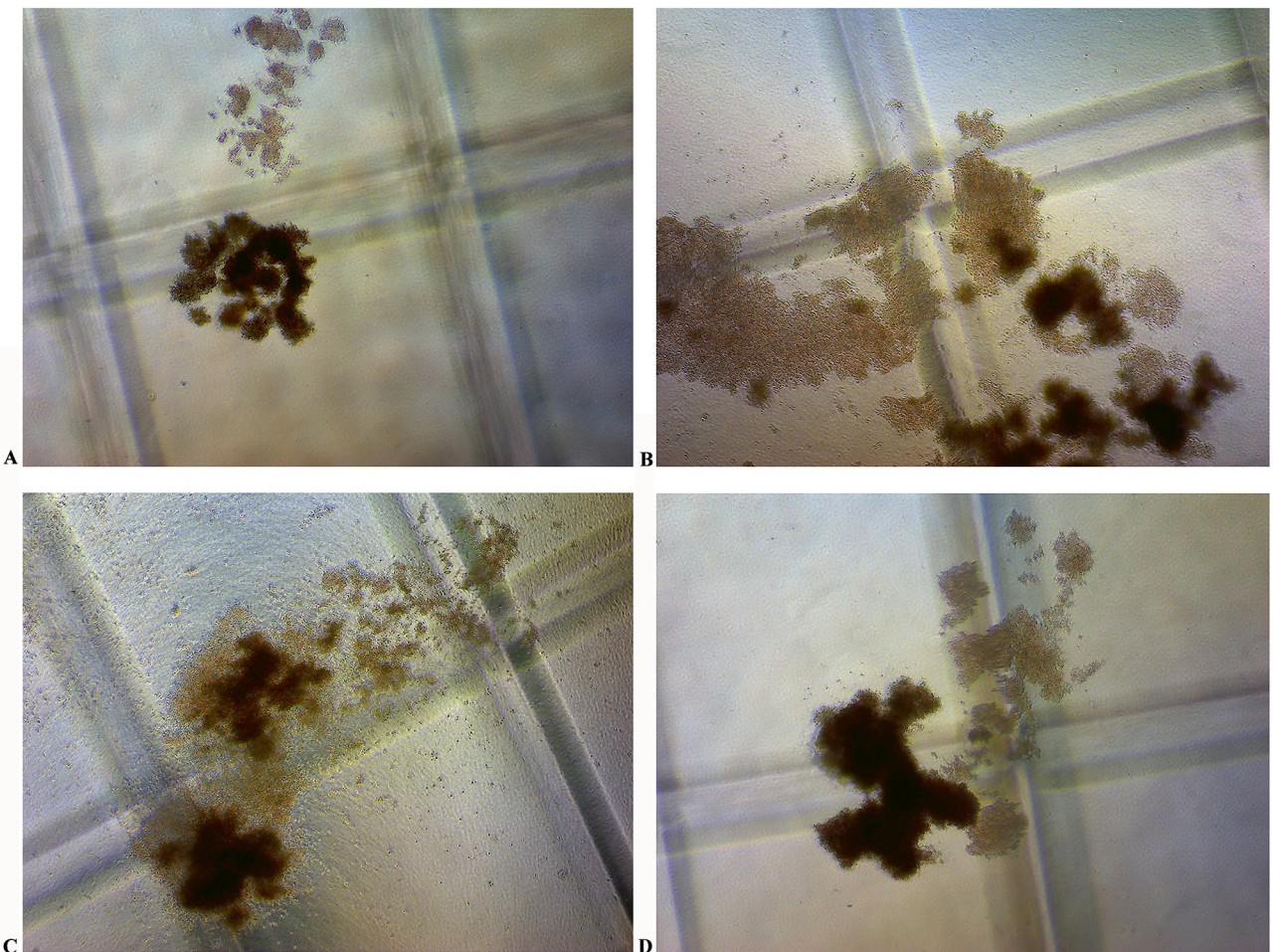

**Fig 4. Effects of lyophilized powder of CBMHL on hematopoietic stem cell colony morphology (40×).** (A) Control Group. (B) CBMHL sample 1. (C) CBMHL sample 2. (D) CBMHL sample 3.

and no animal death occurred during the modeling process. In addition, the red blood cell count, HB, and HCT of rats in the Model Group decreased significantly($P < 0.01$), and the renal function indicators serum BUN and Cr levels increased significantly($P < 0.05$), which were consistent with the characteristics of renal anemia, indicating successful modeling. The results were shown in Table 4.

**Table 3. Effects of lyophilized powder of CBMHL on hematopoietic stem cell colony growth.**

| Group | Number of colonies |
|---|---|
| Control | 51.36±5.00 |
| CBMHL sample 1 | 86.67±2.31* |
| CBMHL sample 2 | 60.33±5.86 |
| CBMHL sample 3 | 55.67±5.69 |

Data expressed as mean ± SD, $n$ = 3.

*$P < 0.05$ *vs.* Control group.

**Table 4. Blood routine and renal function indexes of rats after modeling.**

| Group | RBC (×10¹²/L) | HB (g/L) | HCT (%) | BUN(mmol/L) | Cr (μmol/L) |
|---|---|---|---|---|---|
| Normal | 7.26±0.62 | 164.50±6.16 | 41.50±3.42 | 8.86±1.02 | 57.70±7.48 |
| Model | 4.76±0.57** | 90.38±10.01** | 27.42±3.42** | 13.99±1.17* | 82.00±14.86* |

Data expressed as mean ± SD, $n$ = 6.

*$P$ < 0.05,

**$P$ < 0.01 *vs*. Normal Group.

After CBMHL treatment, the blood routine and renal function indexes of each group were shown in Fig 5. Compared with the Normal Group, the Model Group showed a significant decrease in RBC ($P$ < 0.05), HB and HCT ($P$ < 0.01), and a significant increase in serum BUN and Cr ($P$ < 0.05). Compared with the Model Group, the Model+EPO Group showed a significant increase in RBC ($P$ < 0.05), HB ($P$ < 0.01), and HCT ($P$ < 0.05), with no significant difference in renal function indexes ($P$ > 0.05); the Model+L-CBMHL Group showed no significant improvement in blood routine and renal function indexes ($P$ > 0.05); the Model+-M-CBMHL Group showed a significant increase in HB and HCT ($P$ < 0.05), with no significant improvement in renal function indexes ($P$ > 0.05); the Model+H-CBMHL Group showed a significant increase in RBC, HB, and HCT ($P$ < 0.05), and a significant decrease in serum Cr ($P$ < 0.05). In addition, there were no significant difference in serum BUN/Cr index among all groups ($P$ > 0.05). There was no significant difference between the CBMHL oral groups ($P$ > 0.05), but it could be seen that blood routine tended to increase with the increase of CBMHL dose, while BUN and Cr tended to decrease. The improvement of blood routine in each CBMHL oral group was not as good as that in the Model+EPO Group. These results suggested that CBMHL had the effect of improving erythrocyte hematopoiesis and renal function in rats with renal anemia, and there was a dose-effect relationship,but its erythrocyte hematopoietic effect was not as significant as EPO at this dose.

After CBMHL treatment, compared with the Normal Group, the levels of serum EPO and GSH in the Model Group decreased significantly($P$ < 0.05); compared with the Model Group, the levels of serum EPO and GSH increased significantly in the Model+EPO Group and Model+H-CBMHL Group ($P$ < 0.05); although there was no significant difference between the treatment groups ($P$ > 0.05), but with the increase of oral dose of CBMHL, serum EPO and GSH levels showed an increasing trend. Details were shown in Fig 6. These results suggested that CBMHL could increase the serum levels of EPO and GSH in RA rats, and there was a certain dose relationship.

## 3.5. The effect of CBMHL lyophilized powder on the fragility of rat red blood cells

Red blood cells suspended in isotonic saline can maintain the state of double-sided concave disc. If the osmotic pressure increases, the water of red blood cells will extravasate and the cells shrink. If the osmotic pressure decreases, the water will infiltrate into the cells, causing the red blood cells to expand, rupture and lead to hemolysis. Erythrocyte osmotic fragility test detects the resistance of red blood cells to different concentrations and hypotonic salt solutions. In the experiment, the higher the mass concentration of NaCl corresponding to the beginning and complete hemolysis of red blood cells, the smaller the ability of red blood cells to resist hypotonic salt solution, and the greater the osmotic fragility. On the contrary, the resistance increases and the osmotic fragility decreases.

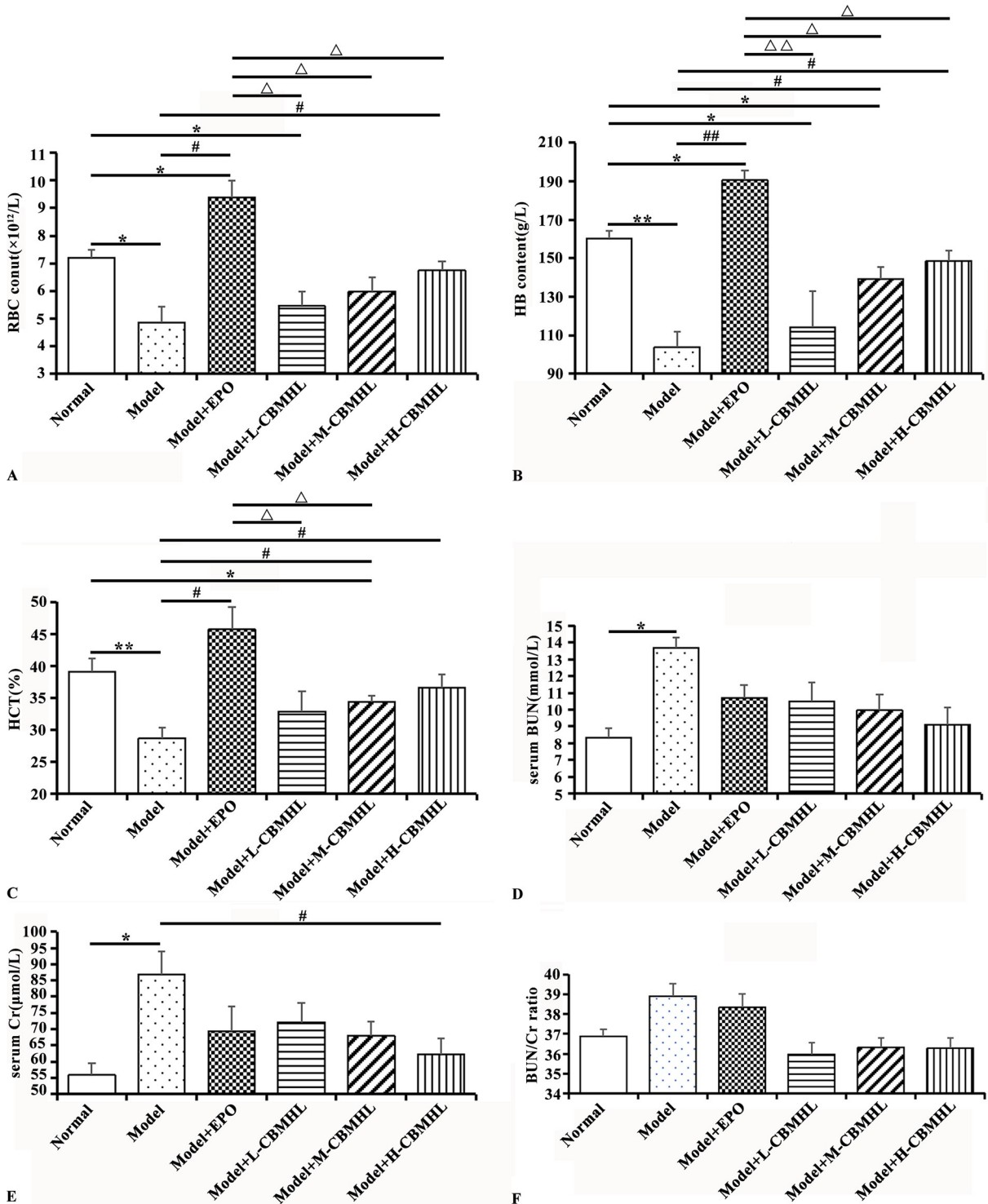

**Fig 5. Effects of lyophilized powder of CBMHL on blood routine and renal function in rats.** (A) RBC count. (B) HB content. (C) HCT. (D) Serum BUN. (E) serum Cr. (F) BUN/Cr ratio. (Data expressed as mean ± SD, $n = 6$; *$P < 0.05$, **$P < 0.01$ *vs*. Normal Group; #$P < 0.05$, ##$P < 0.01$ *vs*. Model Group, $^{\Delta}P < 0.05$, $^{\Delta\Delta}P < 0.01$ *vs*. Model+EPO Group).

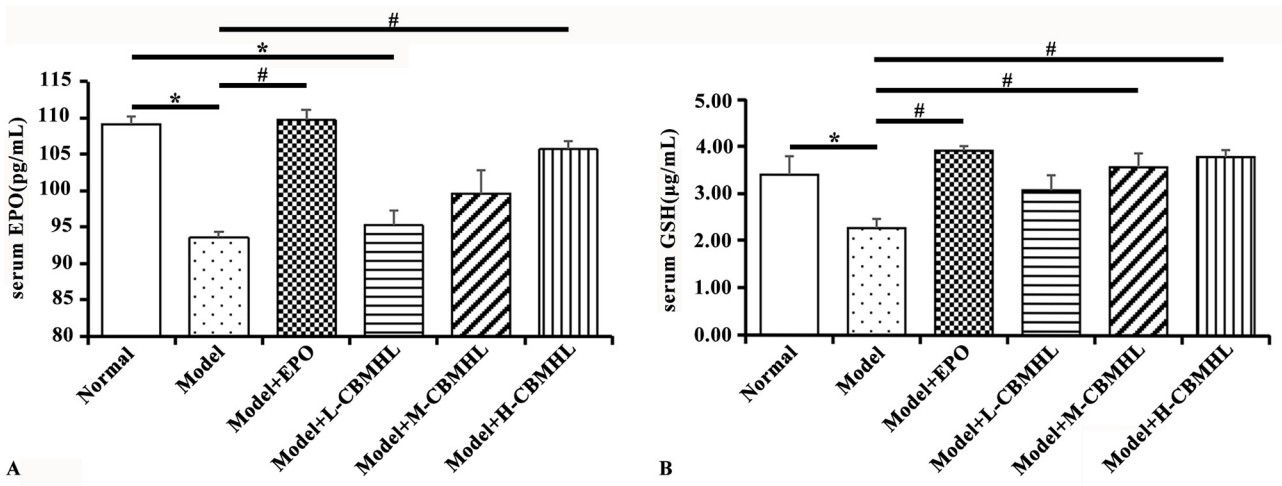

**Fig 6. Effects of lyophilized powder of CBMHL on serum EPO and GSH content in rats.** (A) Serum EPO. (B) Serum GSH. (Data expressed as mean ± SD, $n$ = 6;*$P$ < 0.05, **$P$ < 0.01 *vs*. Normal Group; #$P$ < 0.05, ##$P$ < 0.01 *vs*. Model Group).

After CBMHL treatment, compared with the Normal Group, the NaCl mass concentration at the beginning of hemolysis in the other groups was: Model+EPO Group, Model+L-CBMHL Group, Model+H-CBMHL Group = Normal Group, Model+M-CBMHL Group > Normal Group; the NaCl mass concentration at complete hemolysis was: Model+EPO Group, CBMHL oral groups > Normal Group. Compared with the Model Group, the NaCl mass concentration at the beginning of hemolysis in the other groups was: Model+EPO Group, CBMHL oral groups < Model Group; the NaCl mass concentration at complete hemolysis was: Model+EPO Group, CBMHL oral groups < Model Group. This suggests that the fragility of rat red blood cells in the Model+EPO Group and CBMHL oral groups was improved. Results are shown in Table 5.

## 3.6. The influence of lyophilized powder of CBMHL on the pathological status of rat kidneys

After treatment with CBMHL, the renal pathological results of each group were shown in Fig 7: the Normal Group had abundant and well-formed glomeruli, clear and appropriately sized renal cyst cavities, normal renal tubular structure, and clear interstitial structure; compared

**Table 5. Effect of lyophilized powder of CBMHL on erythrocyte fragility in rats.**

| Group | NaCl mass concentration (g/L) | | | | | | | | | |
|---|---|---|---|---|---|---|---|---|---|---|
| | 5.0 | 4.8 | 4.6 | 4.4 | 4.2 | 4.0 | 3.8 | 3.6 | 3.4 | 3.2 |
| Normal | | | | | + | + | + | + | + | # |
| Model | | | + | # | # | # | # | # | # | # |
| Model+EPO | | | | | + | + | # | # | # | # |
| Model+L-CBMHL | | | | | + | + | # | # | # | # |
| Model+M-CBMHL | | | | + | + | + | + | + | # | # |
| Model+H-CBMHL | | | | | + | + | + | # | # | # |

Data expressed as mean ± SD, $n$ = 6.

+ begining of hemolysis; # complete hemolysis.

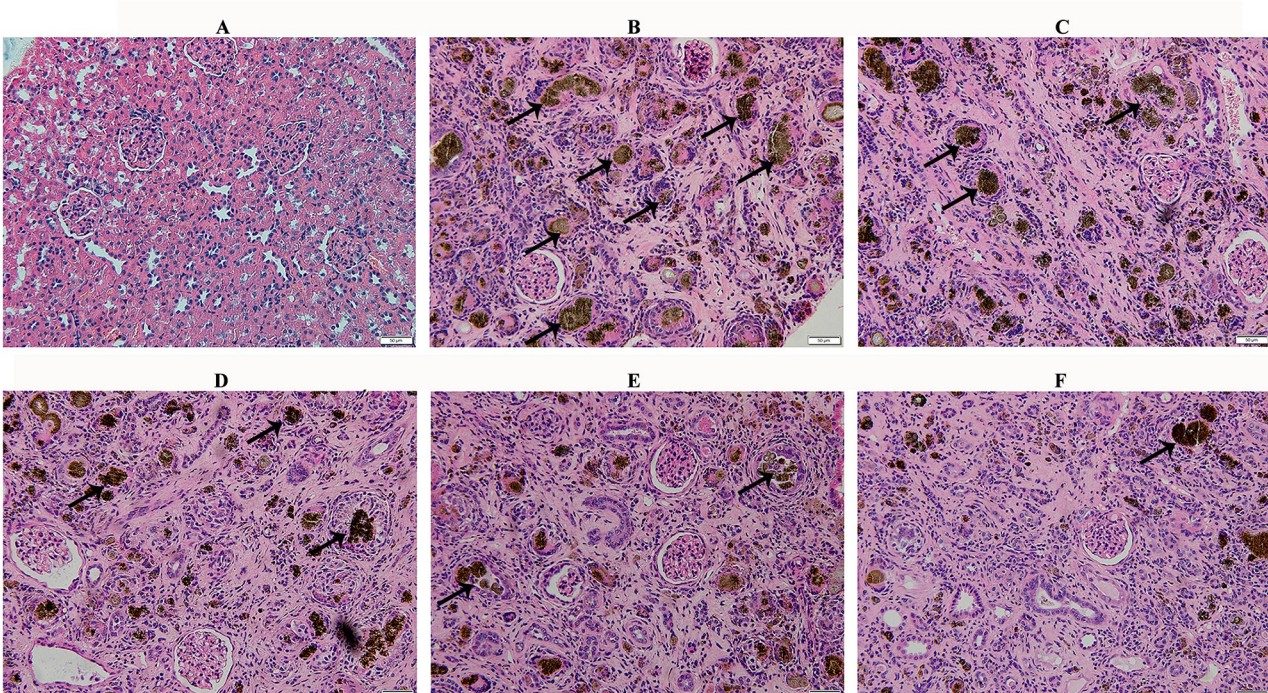

**Fig 7. Effects of lyophilized powder of CBMHL on the pathological morphology of kidney in renal anemia rats (H&E staining, black arrows indicate the deposition of 2,8-DHA crystals, 200×).** (A) Normal Group. (B) Model Group. (C) Model+EPO Group. (D) Model+L-CBMHL Group. (E) Model+M-CBMHL Group. (F) Model+H-CBMHL Group.

with the Normal Group, the number of glomeruli in the Model Group decreased and some glomeruli atrophied, the renal cyst cavities enlarged, brown-yellow crystals and dilated lumen were observed in renal tubules, epithelial cells of the renal tubule proliferated, and there was inflammatory cell infiltration in the interstitium. Compared with the Model Group, the renal pathological morphology of the Model+EPO Group and the CBMHL oral groups improved to varying degrees, especially the Model+H-CBMHL Group showed the most significant improvement, with reduced glomerular atrophy, significantly reduced brownish-yellow crystals in the renal tubules, reduced tubular dilation, and reduced epithelial cell proliferation, and alleviated inflammatory infiltration in the interstitium. In conclusion, CBMHL could significantly improve the pathological morphology of the kidneys.

### 3.7. Lyophilized powder of CBMHL affected the transcription of several hematopoietic factors and receptors in rats

After oral treatment with CBMHL, the transcription levels of several hematopoietic factors and receptors in each group were shown in Fig 8. For the transcription level of renal *Epo*, the Model Group was significantly lower than the Normal Group ($P < 0.01$), the Model+EPO Group, Model+L-CBMHL Group, and the Model+M-CBMHL Group had no signifiant difference compared with the Model Group respectively ($P > 0.05$), while the Model+H-CBMHL Group was significantly higher than the Model Group ($P < 0.05$). For the transcription level of *Epor* in bone marrow, the Model Group was significantly lower than the Normal Group ($P < 0.05$), there was no significant difference between the Model+L-CBMHL Group and the Model Group ($P > 0.05$), while the Model+EPO Group, Model+M-CBMHL Group, and

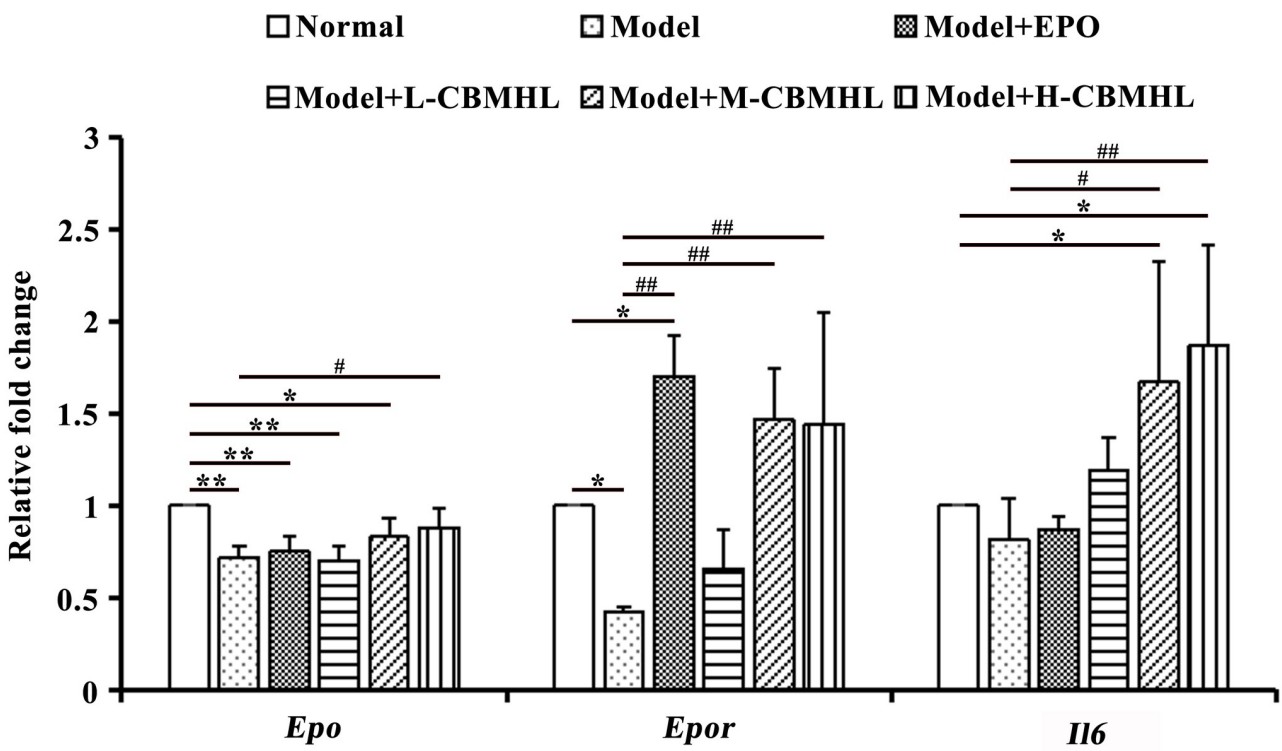

**Fig 8. Effects of lyophilized powder of CBMHL on transcription of some hematopoietic factors and receptors in rats.** (Data expressed as mean ± SD, $n = 6$; *$P < 0.05$, **$P < 0.01$ *vs*. Normal Group; #$P < 0.05$, ##$P < 0.01$ *vs*. Model Group).

Model+H-CBMHL Group were significantly higher than the Model Group ($P < 0.01$). For the transcription level of *Il6* in spleen tissue, compared with the Normal Group, there was no significant difference in the Model Group, Model+EPO Group, and Model+L-CBMHL Group ($P > 0.05$), while the Model+M-CBMHL Group and Model+H-CBMHL Group were significantly higher ($P < 0.05$); compared with the Model Group, there was no significant difference in the Model+EPO Group and Model+L-CBMHL Group ($P > 0.05$), while the Model+-M-CBMHL Group was significantly higher ($P < 0.05$), and the Model+H-CBMHL Group was also significantly higher ($P < 0.01$). In summary, the above results suggested that CBMHL could promote hematopoiesis by promoting the expression of renal *Epo*, bone marrow *Epor*, and spleen *Il6* in rats.

## 4. Discussion

At present, enzymatically hydrolyzed protein is an important source of food derived active polypeptides [14]. In this experiment, newborn bovine bone marrow with high blood production function was used as raw material. After low-density degreasing and papain hydrolysis (degreasing rate 99.5%, protein hydrolysis rate 75.70%), the peptide form of calf bone marrow hydrolysate was obtained. After research, the peptide form of amino acids is more easily absorbed and utilized by cells, and has potential biological activity [13]. As a commonly used drug loading preparation, liposomes encapsulate drugs in lipid bilayer to form ultramicro vesicles. Their structure and characteristics are similar to those of biofilms. After entering the body, they are easy to fuse, adsorb and exchange lipids with intestinal mucosal cells, enter the systemic circulation through the intestinal epithelium, and protect them from degradation by

enzymes in the stomach and intestine, so as to improve the bioavailability [15]. Considering the above advantages of liposomes, the traditional thin-film hydration technology was used in this experiment to prepare calf bone marrow hydrolysate liposomes (the mass ratio of soybean lecithin to cholesterol was 1:3, and the mass ratio of calf bone marrow hydrolysate to soybean lecithin was 1:15). Then the key parameters of liposomes such as appearance, particle size, dispersity, encapsulation efficiency and morphology were characterized, which was basically consistent with the characteristics of liposomes in references [19]. Ordinary liposomes are liquid preparations, which are prone to particle aggregation and sedimentation, leakage of encapsulated drugs and other problems, resulting in unstable liposomes [16, 21]. Therefore, in order to improve the storage stability of liposomes, they are often made into lyophilized powder. In this experiment, mannitol, a commonly used freeze-drying protectant with a mass volume ratio of 5%, was added to the calf bone marrow hydrolysate liposomes, and a lyophilizer was used to adjust the temperature and pressure changes. The liposomes were freeze-dried into powder through pre-freezing, sublimation drying and analytical drying. The powder was plump and uniform in appearance, with no color difference from top to bottom; after adding pure water, it could dissolve rapidly, the rehydration time was less than 2 s, and the color of the dissolved liquid was homogeneous, suggesting that the lyophilized powder of liposomes had good redispersity, and CBMHL were easier to recover activity after rehydration; the water content directly affected the storage time of freeze-dried powder. With high water content, bacteria and fungi are more likely to breed. According to the measurement, the water content of freeze-dried powder was < 1.8%, which was low and the powder was easy to store; after storaged at 4 ˚C for 8 weeks, compared with the initial liquid liposomes, the average particle size, PDI and EE% of freeze-dried liposomes did not change significantly, suggesting that freeze-drying technology indeed enhanced the stability of liposomes. In conclusion, from the appearance, redispersibility, water content, liposome indexes and stability of the freeze-dried powder, it could be concluded that the quality of freeze-dried powder of CBMHL under this freeze-drying process was good, and the production process was relatively simple, which provided a basis and reference for the future possible industrial production.

Adipose MSCs are derived from adipose tissue, which is widely distributed in the adult body, such as bone marrow, joints, subcutaneous and internal organs [22]. Recent studies have found that adipose MSCs play an important biological role in the body. Firstly, they have the ability to differentiate into mesodermal-derived cells, including adipocytes, chondrocytes, osteoblasts, etc., and timely update and repair damaged cells or tissues. Secondly, they secrete or produce various cytokines, growth factors, nucleic acids (miRNA) and other macromolecules into the surrounding environment through microvesicls, to change tissue biology, stimulate tissue resident stem cells, and alter immune cell activity [28]. This experiment co-cultured adiposed MSCs with lyophilized powder of CBMHL to observe the apoptosis of MSCs, in order to study whether there was potential toxicity of freeze-dried powder of CBMHL on human adipose MSCs in vitro. The experimental results showed that there was no significant difference in the apoptosis rate of MSCs in each group, suggesting that lyophilized powder of CBMHL had no potential toxicity to human adipose MSCs. Food is digested and absorbed by the gastrointestinal tract and enters the blood circulation. If certain factors in the food can stimulate the body's immune system and break the immune balance, it often causes an immune response in the body. CIK cells are a heterogeneous group of cells induced in vitro by various cytokines [23]. They exist naturally in the body and regulate the immune activity of the body by releasing toxic particles such as granzyme/perforin and releasing a large amount of inflammatory cytokines [29]. In this experiment, CIK cells were induced and cultured in vitro, and co-cultured with different concentrations of lyophilized powder of CBMHL to observe whether the proliferation ability of CIK cells changed, in order to evaluate whether the

freeze-dried powder of CBMHL caused allergy in the body. The experimental results showed that there was no significant difference in the proliferation of CIK cells in each group after co-culturing for 12h, 24h, 36h, and 48h, suggesting that the freeze-dried powder of CBMHL generally did not cause allergic reactions in the body.

In vitro, we observed the effect of CBMHL on hematopoietic stem/progenitor cells through the typical cell semi-solid culture technology [30], to evaluate whether it had the effect of promoting hematopoiesis. Hematopoietic stem cells (HSCs) are a unique group of cells that can differentiate into mature blood cells and can also self-renew [31]. Currently, hematopoietic stem cells can be isolated from bone marrow, umbilical cord blood, and peripheral blood [32]. In this study, cord blood MNCs were isolated and cultured in an appropriate semi-solid medium. The characteristics of individual progenitor cells proliferating and differentiating into discrete colonies were observed, and the number of colonies was statistically analyzed to determine whether CBMHL could promote hematopoietic function. In this experiment, after continuous culture of cord blood hematopoietic stem cells for 14 days, compared with the control group, the volume of BFU-E colonies in CBMHL samples 1, 2, and 3 increased significantly, and the number of colonies in CBMHL sample 1 increased significantly, indicating that CBMHL can indeed promote the differentiation and proliferation of hematopoietic stem cells into early red cell progenitor cells *in vitro*. It was speculated that after oral administration of lyophilized powder of CBMHL, hematopoietic active substances were absorbed by intestinal epithelial cells, circulated in the blood to various parts of the body, promoted the differentiation and proliferation of hematopoietic stem cells into early red cell progenitor cells, and further proliferated and differentiated into mature red blood cells, improving the symptoms of anemia. However, whether this speculation was true needed to be further verified through animal experiments.

In this study, a renal anemia model was established in rats by oral administration of adenine, and CBMHL was fed to observe its auxiliary hematopoietic function *in vivo*. The main methods for establishing a renal anemia animal model include drug induction, 5/6 nephrectomy, and unilateral ureteral obstruction [33, 34]. Among them, adenine-induced renal anemia model is the most widely used. According to the research, adenine is a substance with obvious toxicity to the kidney, high concentrations of adenine in the body metabolize to form 2,8-DHA crystals, a brownish-yellow, highly insoluble crystalline substance that blocks renal tubules, leading to the accumulation of various guanidine compounds and the increase of bun and Cr concentrations, causing great damage to the kidney structure and leading to renal failure and anemia [35]. In this experiment, after 28 days of adenine administration, compared with the Normal Group, the rats in the Model Group showed rough and dull fur, hair loss, emaciation, hunchback, huddling, lethargy, and increased urine output. In addition, blood routine indexes (RBC count, Hb content, Hct) significantly increased, and renal function indexes (BUN, Cr) significantly increased. The above results were basically consistent with the relevant descriptions in the references, which verified that the renal anemia model was successfully established [36]. After continuous oral treatment of CBMHL for 21 days, compared with the Model group, especially the Model+H-CBMHL group, the blood routine indexes (RBC count, Hb content, HCT) were significantly increased, the renal function indexes (Cr) were significantly decreased, and the renal histopathology was also significantly improved. According to the serum BUN and Cr concentrations in the references on renal anemia, the calculated BUN/Cr ratio in each group varied greatly, with some being higher, some lower, and some unchanged compared to the normal group [26, 36–38]. There was no significant difference in the serum BUN/Cr ratio between the groups in this study, which was also consistent with the description in the reference. Considering the above situation, it was likely that this value could be easily affected by various factors other than the kidney. The above results suggested that

CBMHL could remedy anemia symptoms, improved renal function and reduced the pathological morphology of kidney tissue to a certain extent, and there was a dose effect relationship. Although the ability of CBMHL to promote hematopoiesis at this dose was not as good as EPO, it still had certain advantages considering that it was derived from food and has relatively high safety.

In adults, 90% of erythropoietin (EPO) is synthesized and secreted by the renal cortex [34]. EPO binds to its receptor (EPOR) to produce various effects, such as promoting red blood cell differentiation and maturation, activating the phosphoinositide 3-kinase/protein kinase B (PI3K/AKT) signaling pathway to protect renal tissue, and exerting antioxidant stress on non-hematopoietic tissues [39]. EPO also enhances iron absorption by the bone marrow [39]. Kidney injury leads to decreased EPO levels, resulting in anemia, and continued ischemia worsens kidney function [38]. Currently, subcutaneous injection of EPO is the main method for treating renal anemia, and iron supplements can also be used as an adjunct [39]. In the renal anemia model, impaired renal metabolic function leads to the accumulation of toxic substances in the body, which disrupts the hematopoietic microenvironment of the bone marrow and inhibits the expression of EPOR in hematopoietic cells, exacerbating anemia [34]. In this experiment, the serum EPO content of Model group rats was significantly reduced, and the transcription levels of renal EPO and bone marrow EPOR reduced significantly, which was consistent with the previous references. After 21 days of continuous oral treatment with CBMHL, especially in the Model+H-CBMHL Group, serum EPO content and the transcription levels of *Epo* in the kidney and *Epor* in the bone marrow increased significantly. In addition, the comparison between CBMHL oral groups indicates that with the increase of oral dose, the content of serum EPO and the transcription levels of kidney EPO and bone marrow EPOR of rats increased. These results suggested that the hematopoietic active substances in CBMHL freeze-dried powder might upregulate the expression of the renal *Epo* gene, promote renal EPO secretion, upregulate the expression of *Epor* gene in hematopoietic cells of the bone marrow, enhanced the reactivity between EPO and EPOR in nucleated cells of the bone marrow, and produced various effects, such as promoting erythropoiesis in the bone marrow and activating the PI3K/AKT signaling pathway to enhance the protective effect of the kidney.

A large number of studies have shown that toxic substances such as arginine in the plasma of the renal anemia model can affect the function of the Na pump on the red blood cell membrane, leading to increased red blood cell fragility and reduced red blood cell survival time [40, 41]. In addition, kidney diseases are often accompanied by oxidative stress, which increases the production of oxygen free radicals in the body, promotes lipid peroxidation of cell membranes, further increases red blood cell fragility, shortens their lifespan, and exacerbates the symptoms of anemia [26, 42]. Glutathione (GSH) can scavenge free radicals and alleviate the damage caused by excessive free radicals to tissues and cells [43]. After CBMHL treatment, the serum GSH levels in the Model+M-CBMHL Group and Model+H-CBMHL Group increased significantly, and the red blood cell fragility in the CBMHL oral groups improved to some extent, indicating that CBMHL could improve the symptoms of anemia by enhancing the function of the body's free radical scavenging system and reducing red blood cell fragility.

Research shows that when the bone marrow microenvironment is disrupted and hematopoietic function is reduced, the spleen compensates by undergoing extramedullary hematopoiesis [44]. IL6, as an important hematopoietic regulatory factor, has multiple effects and can collaborate with other hematopoietic factors such as EPO, granulocyte-macrophage colony stimulating factor (GM-CSF), interleukin 3 (IL3), and stem cell stimulating factor (SCF) to promote hematopoietic function recovery [45]. After CBMHL treatment, the transcription level of *Il6* in the spleen tissue of rats in the Model+M-CBMHL Group and Model+H-CBMHL

Group significantly increased, indicating that CBMHL might promote hematopoiesis by upregulating the expression of *Il6* gene in the spleen.

## 5. Conclusion

This study demonstrated that CBMHL could promote bone marrow hematopoiesis by upregulating the expression of *Epo* gene in the kidneys, promoting EPO secretion in the kidneys, and upregulating the expression of *Epor* gene in the bone marrow. It significantly improved various red blood cell hematopoietic indicators in anemia model rats, and significantly increased the expression of major hematopoietic factors in the body. It also significantly enhanced the function of the free radical clearance system, reduced red blood cell fragility, and had the function of promoting hematopoiesis and reducing anemia. At the same time, it significantly improved the pathological morphology and function of the kidneys. In summary, oral administration of CBMHL lyophilized powder significantly improved anemia symptoms both in vitro and in vivo. This study opened up new ideas and provided scientific basis for the development of new blood supplement products.

## Supporting information

**S1 Fig. Certificate for laboratory animal.**
(TIF)

**S2 Fig. The appearance and redispersability of lyophilized powder of CBMHL.** (A) Appearance. (B) Redispersability.
(TIF)

**S1 File. Ethical approval for cord blood collection.**
(PDF)

**S2 File. Ethical approval for animal.**
(PDF)

## Acknowledgments

Authors are thankful to the Shandong University by supporting this research through providing experimental materials and instruments.

## Author Contributions

**Conceptualization:** Li Li, Dong Li.

**Data curation:** Li Li, Dong Li.

**Formal analysis:** Li Li.

**Funding acquisition:** Shasha Zhao, Xiaoyu Dai.

**Investigation:** Shasha Zhao, Zhe Xu.

**Methodology:** Li Li, Shasha Zhao.

**Project administration:** Xiaoyu Dai.

**Resources:** Shasha Zhao, Dong Li.

**Supervision:** Dong Li, Xiaoyu Dai.

**Writing – original draft:** Li Li, Xiaodun Liu, Dong Li.

**Writing – review & editing:** Li Li, Xiaodun Liu, Dong Li, Xiaoyu Dai.

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
