## [Decision Letter · Decision Letter 0]

27 Sep 2024

PONE-D-24-31910Lyophilized powder of calf bone marrow hydrolysate liposomes improved renal anemia: In vitro and in vivo evaluationPLOS ONE

Dear Dr. Dai,

Thank you for submitting your manuscript to PLOS ONE. After careful consideration, we feel that it has merit but does not fully meet PLOS ONE’s publication criteria as it currently stands. Therefore, we invite you to submit a revised version of the manuscript that addresses the points raised during the review process.

The manuscript of Li et al. reported the effect of CBMHL in renal anemia developed rat model. The manuscript needs significant improvement as suggested by the reviewers. Please address all the academic editor and reviewers comments. 

Academic Editor Comments

Please characterize the synthesized lysosome properly (also suggested by the reviewer 2)The level of creatinine and blood urea nitrogen is higher than the normal reference range. In the discussion section, please mention why their level is high in the control group. Additionally, please compare the results with recent literature of the same rat species. Please convert the result of BUN and creatinine in the same unit, calculate BUN to creatinine ratio and compare the findings. You can get relevant information here [https://pubs.acs.org/doi/full/10.1021/acsabm.0c01069]. Please report the result of Table 4, 5 in bar diagram format and also perform multiple comparison between the treatment groups using one-way Anova coupled with a suitable post-hoc test. Re-write the result and discussion section accordingly. 

We look forward to receiving your revised manuscript.

Kind regards,

Salman Shakil

Academic Editor

PLOS ONE

“This work was supported by Shandong Province Key R&D Program (Medical Food Special Program) Project (No.2019YYSP009).”

Reviewers' comments:

Reviewer's Responses to Questions

**Comments to the Author**

1. Is the manuscript technically sound, and do the data support the conclusions?

Reviewer #1: Partly

Reviewer #2: Yes

2. Has the statistical analysis been performed appropriately and rigorously? 

Reviewer #1: Yes

Reviewer #2: Yes

3. Have the authors made all data underlying the findings in their manuscript fully available?

Reviewer #1: Yes

Reviewer #2: Yes

4. Is the manuscript presented in an intelligible fashion and written in standard English?

Reviewer #1: Yes

Reviewer #2: Yes

5. Review Comments to the Author

Reviewer #1: 1. Page 9, Line 164-165: Model+L-CBMHL, Model+M-CBMHL, and Model+H-CBMHL have been abbreviated. The meaning of L, M, and H are not defined. At first appearance, authors should define what L, M, and H are?

2. The authors used “Rat” as an animal model. The type and species of the rat should be mentioned.

3. In 2.2.2 the author described the preparation process of liposomes. The soybean lecithin and cholesterol mass ratio is 3:1 but the authors did not mention the amount/ratio of CBMHL in the preparation. The authors should mention it.

4. The reported particle size is 50 � 30 nm. For 50 nm size particles, a standard deviation of 30 nm is very high. Do you have any explanation for this? And, what about the polydispersity index of particle size measurement?

5. In Figure 2, it seems that the author represented a single measurement data instead of an average. I recommend presenting a graph of average data.

6. On page 4, line 73, the authors said that liposomes produced freeze-drying process enhance stability. But, throughout the article, there is no data or experiment about the stability of CBMHL-containing liposomes. The authors should carry out the stability study of liposomes.

7. The authors characterized liposomes only based on particle size, is there any other vital parameter to characterize liposomes?

8. On page 18, line 334 the authors said that “the CBMHL freeze-dried powder has good quality and……” it should be mentioned that based on what data the quality of the powder is good.

9. There is no information about the toxicity of the formulation (in vitro and in vivo). At least in vitro toxicity studies should be carried out and represented.

Reviewer #2: The manuscript titled "Lyophilized powder of calf bone marrow hydrolysate liposomes improved renal anemia: In vitro and in vivo evaluation" is suitable for publication in the PLOS ONE. However, it must undergo modifications and address both major and minor corrections before it meets the criteria for publication in the mentioned journal.

I suggest approving this manuscript once the following revisions are completed.

Major comments

1. In methods section, you didn’t mention any reference(s). Did you perform all methods as first time? You should provide references in this section if necessary.

2. I noticed many grammatical errors through the manuscript. Need to update carefully.

a. Specially the singular and plural form.

b. Active and passive format

c. Mixed with present and past tense in same sentence or on same topics.

3. How you confirmed that the synthesis nanoparticle is liposome not others nanoparticle like LNP? You characterized only by particle size analyzer but didn’t characterize by others analytical tools like TEM or SEM. Additionally you only provide the information for particle size, not PDI, peak intensity (you can refer this article as an example

https://pubs.acs.org/doi/10.1021/acsabm.2c00061

https://doi.org/10.3390/molecules28072969 )

4. You didn’t show any stability data of your formulations. Is there any size differentiation over the time through nano-zetasizer experiments? Is there any size different between drug incorporation and without drugs molecules in liposome? At ease 1 month’s stability data is recommended. (you can use this article as references for formulation stability:

https://pubs.acs.org/doi/10.1021/acsabm.1c00563

5. In discussion section, there is lack of scientific and logical explanation based on your research outcome. You mentioned many descriptive information but not sufficient supportive scientific explanations for your research outcome in results section. Please improve the discussion section to support your result.

Minor comments

1. In the descriptive sections, you don’t need to mark with identical pools (such as triangle, square, star, or any others marked symbols). In should be as it is that’s are align with previous text or paragraph (line 163-165).

2. How you select dose 250 mg/kg/d? Is there any reference value for this dosage form to get proper pharmacokinetic response? (Line 155)

3. In figure-4, measuring scale is not visualized. Need to make it clear.

4. Funding information is missing.

5. Declaration or conflict of interest is missing.

6. PLOS authors have the option to publish the peer review history of their article (what does this mean?). If published, this will include your full peer review and any attached files.

Reviewer #1: No

Reviewer #2: **Yes: **SHIHAB UDDIN

---

## [Author Response · Author response to Decision Letter 0]

11 Nov 2024

Dear editors and reviewers:

Firstly, thank you for your review and valuable feedback on this article. We have carefully revised and supplemented the paper according to your feedback, and all the modifications have been marked in different colored fonts in the revised draft.

For the attached revision suggestions, we have made amendments and supplements item by item as follows:

Academic Editor Comments

1. Please characterize the synthesized lysosome properly (also suggested by the reviewer 2).

Thank you for the suggestion. We have recharacterized the synthesized liposomes, including appearance, particle size, dispersity, encapsulation efficiency, and morphology. The original was amended as follows, the synthesized liposome was white, bluish, and translucent emulsion by naked eye observation; The average particle size of liposomes was (46.98 ± 0.67) nm and PDI was 0.235 ± 0.011, as detected by a nanoparticle analyzer; Using UV spectrophotometry to detect the concentration of peptides inside and outside the liposome, calculate the encapsulation efficiency of the liposome, which was (93.13 ± 0.38)%; By observing the morphology of liposomes using TEM, it appeared as a uniform spherical or quasi spherical shape, and is a type of microbubble with lipid bilayer.

2. The level of creatinine and blood urea nitrogen is higher than the normal reference range. In the discussion section, please mention why their level is high in the control group. Additionally, please compare the results with recent literature of the same rat species.

Thank you for your question. In response to your question, we made a supplementary explanation in the discussion section, and compared the results with the recent literature on Wistar rats.

3. Please convert the result of BUN and creatinine in the same unit, calculate BUN to creatinine ratio and compare the findings. You can get relevant information here [https://pubs.acs.org/doi/full/10.1021/acsabm.0c01069].

Thank you for your question. The original was amended as follows, according to the serum BUN and Cr concentrations in the references on renal anemia, the calculated BUN/Cr ratio in each group varied greatly, with some being higher, some lower, and some unchanged compared to the normal group[26, 36-38]. There was no significant difference in the serum BUN/Cr ratio between the groups in this study, which was also consistent with the description in the reference. Considering the above situation, it was likely that this value could be easily affected by various factors other than the kidney.

4. Please report the result of Table 4, 5 in bar diagram format and also perform multiple comparison between the treatment groups using one-way Anova coupled with a suitable post-hoc test. Re-write the result and discussion section accordingly.

Thank you for your suggestion. We showed the results of Table 4 and 5 in a graphical format. In addition, we used one-way analysis of variance (ANOVA) to detect the overall differences between groups firstly, and used LSD method to make multiple comparisons between treatment groups secondly, lastly rewrote the results and discussion section.

Reviewer #1:

1.Page 9, Line 164-165: Model+L-CBMHL, Model+M-CBMHL, and Model+H-CBMHL have been abbreviated. The meaning of L, M, and H are not defined. At first appearance, authors should define what L, M, and H are?

Thank you very much for your suggestion. In the original text, we have defined and revised L, M and H. The original was amended as follows, all the rats in the experiment were grouped as follows: Normal Group, Model Group, Model+EPO Group, CBMHL oral groups (Model+low doses of CBMHL Group (Model+L-CBMHL Group), Model+medium doses of CBMHL Group (Model+M-CBMHL Group), Model+high doses of CBMHL Group (Model+H-CBMHL Group)).

2. The authors used “Rat” as an animal model. The type and species of the rat should be mentioned.

Thank you for your question. This experiment used an animal model made by Wistar rats, which was mentioned in the materials and instruments section of the article (line 91).

3. In 2.2.2 the author described the preparation process of liposomes. The soybean lecithin and cholesterol mass ratio is 3:1 but the authors did not mention the amount/ratio of CBMHL in the preparation. The authors should mention it.

Thank you for your question. According to your suggestion, we have added this part to the article, in which the mass ratio of calf bone marrow hydrolysate to soybean lecithin was 1:15. The original was amended as follows, The calf bone marrow hydrolysate-PBS buffer was heated in a constant temperature water bath at 45 ℃ and was poured into the rotary evaporator containing the lipid film (calf bone marrow hydrolysate: soybean lecithin, 1:15 mass ratio).

4. The reported particle size is 50 ± 30 nm. For 50 nm size particles, a standard deviation of 30 nm is very high. Do you have any explanation for this? And, what about the polydispersity index of particle size measurement?

Thank you very much for asking this question. The description of the particle size of the prepared liposomes in the original text was not very accurate, and there were misunderstandings, so we recharacterized the liposomes, including appearance, particle size, dispersity, encapsulation efficiency and morphology. The original was amended as follows. According to the naked eye observation, the synthesized liposomes were white, bluish and translucent emulsion (Fig 2A); the average particle size of liposomes was (46.98 ± 0.67) nm and the PDI was 0.235 ± 0.011 detected by nanoparticle size analyzer (Fig 2C and Table 2); the concentration of polypeptide inside and outside the liposomes was detected by UV spectrophotometry, and the encapsulation efficiency was calculated, the encapsulation efficiency was (93.13 ± 0.38)% (Table 2); TEM was used to observe the morphology of liposomes, which were uniform spherical or quasi spherical, and were a kind of micro vesicles with lipid bilayer (Fig 2B).

5. In Figure 2, it seems that the author represented a single measurement data instead of an average. I recommend presenting a graph of average data.

Thank you for your suggestion. The data of particle size and dispersity of liposomes were expressed as average values and shown in Table 2.

Table2. The particle size , PDI and EE% of the different liposome samples.

Name of the sample Particle size(nm) PDI EE%

Initial liposomes 46.98±0.67 0.235±0.011 93.13±0.38

Liposomes after 8 weeks 68.83±1.14** 0.398±0.018** 86.65±0.92**

liposome lyophilized powder after 8 weeks 47.12±0.22 0.206±0.007 92.26±0.58

Data expressed as mean ± SD, n = 3.

**P < 0.01 vs. Initial liposomes.

6. On page 4, line 73, the authors said that liposomes produced freeze-drying process enhance stability. But, throughout the article, there is no data or experiment about the stability of CBMHL-containing liposomes. The authors should carry out the stability study of liposomes.

Thank you for pointing out. According to your suggestion, we added the research experiment and results of freeze-drying technology on the stability of liposomes in the article (line 146-150, line 260-263).

7. The authors characterized liposomes only based on particle size, is there any other vital parameter to characterize liposomes?

According to your suggestion, we recharacterized the liposomes. In addition to the particle size, we added important parameters such as appearance, dispersity, encapsulation efficiency and morphology (line 248-255).

8. On page 18, line 334 the authors said that “the CBMHL freeze-dried powder has good quality and……” it should be mentioned that based on what data the quality of the powder is good.

Thank you for your question. The lyophilized powder of CBMHL liposome had good appearance, plump and uniform and there was no color difference from top to bottom; The lyophilized powder generally presented a loose and porous structure, the rehydration time was less than 2 s, and the liquid color of the solution was uniform, indicating that the lyophilized powder had good redispersibility, and CBMHL was easier to recover its activity after rehydration; Water content directly affectes the storage time of freeze-dried powder. High water content is easy to breed bacteria and fungi and reduced the stability of liposomes. The measured water content of freeze-dried powder was < 1.8%, indicating that the water content was low and easy to store; After storage at 4 ℃ for 8 weeks, compared with liquid liposomes, the average particle size, PDI and encapsulation efficiency of lyophilized powder had no obvious changes, indicating that CBMHL liposomes were of good quality and easier to store after freeze-drying.

9.There is no information about the toxicity of the formulation (in vitro and in vivo). At least in vitro toxicity studies should be carried out and represented.

Thank you for your suggestion. In vitro toxicity test and data were added in the paper (line 151-167, 268-281).

Reviewer #2: The manuscript titled "Lyophilized powder of calf bone marrow hydrolysate liposomes improved renal anemia: In vitro and in vivo evaluation" is suitable for publication in the PLOS ONE. However, it must undergo modifications and address both major and minor corrections before it meets the criteria for publication in the mentioned journal.

I suggest approving this manuscript once the following revisions are completed.

Major comments

1. In methods section, you didn’t mention any reference(s). Did you perform all methods as first time? You should provide references in this section if necessary. 

Thank you for your suggestions. We have added references to the methods section.

2. I noticed many grammatical errors through the manuscript. Need to update carefully.

a. Specially the singular and plural form.

b. Active and passive format

c. Mixed with present and past tense in same sentence or on same topics.

Thank you for your question. We have revised the whole article.

3. How you confirmed that the synthesis nanoparticle is liposome not others nanoparticle like LNP? You characterized only by particle size analyzer but didn’t characterize by others analytical tools like TEM or SEM. Additionally you only provide the information for particle size, not PDI, peak intensity (you can refer this article as an example )https://pubs.acs.org/doi/10.1021/acsabm.2c00061

https://doi.org/10.3390/molecules28072969 )

Thank you for the suggestion. We characterized the liposomes from the aspects of appearance, particle size, dispersity, encapsulation efficiency and morphology. The original text was modified as follows. After visual observation, the synthesized liposomes appeared as white, bluish light and translucent emulsion; The average particle size of liposomes was (46.98 ± 0.67) nm and the PDI was 0.235 ± 0.011 by nanoparticle size analyzer; The concentration of polypeptide inside and outside the liposomes was detected by UV spectrophotometry, and the encapsulation efficiency of liposomes was calculated, which was (93.13 ± 0.38)%; TEM was used to observe the morphology of liposomes, which were uniform spherical or quasi spherical, and were a kind of micro vesicles with lipid bilayer.

4. You didn’t show any stability data of your formulations. Is there any size differentiation over the time through nano-zetasizer experiments? Is there any size different between drug incorporation and without drugs molecules in liposome? At ease 1 month’s stability data is recommended. (you can use this article as references for formulation stability:https://pubs.acs.org/doi/10.1021/acsabm.1c00563

Thank you for your question. In the article, we added the stability study of freeze-dried CBMHL liposomes. The liquid liposomes and freeze-dried liposomes were stored at 4 ℃ for 8 weeks, and then taken out. The average particle size, PDI and encapsulation efficiency of the liposomes were detected respectively. The experimental results showed that compared with the initial liposomes, the average particle size and PDI increased significantly (P < 0.01), encapsulation efficiency of liquid liposomes decreased significantly (P < 0.01), while the average particle size, PDI and encapsulation efficiency of freeze-dried liposomes had no significant change, indicating that although the stability of liquid liposomes in the environment of 4 ℃ was not good under this formula, while storage after freeze-drying process showed a certain degree of stability.

5. In discussion section, there is lack of scientific and logical explanation based on your research outcome. You mentioned many descriptive information but not sufficient supportive scientific explanations for your research outcome in results section. Please improve the discussion section to support your result.

Thank you for your suggestions. According to your suggestions, we have made corresponding modifications and supplements to the results and discussion. 

Minor comments

1. In the descriptive sections, you don’t need to mark with identical pools (such as triangle, square, star, or any others marked symbols). In should be as it is that’s are align with previous text or paragraph (line 163-165).

Thank you very much for your suggestion. We have deleted and modified this part and carefully checked it in the full text.

2. How you select dose 250 mg/kg/d? Is there any reference value for this dosage form to get proper pharmacokinetic response? (Line 155)

Thank you for your question. At present, the main methods of establishing animal models of renal anemia include drug induction, 5/6 nephrectomy, and unilateral ureteral obstruction. Among them, the establishment of renal anemia models by intragastric administration of adenine is most widely used. In this paper, the adenine dose of 250mg/kg/d in rats was selected referring to two literatures [25,26]. We chosed a relatively middle value. According to the research, high concentration of adenine formed water-insoluble 2,8-dihydroxyadenine through the action of xanthine oxidase in the liver, and deposited in renal tubules after glomerular filtration, blocking renal tubule lumen, which caused renal dysfunction due to renal interstitial tubule injury, and easily leaded to renal anemia.

3. In figure-4, measuring scale is not visualized. Need to make it clear.

Thank you for your suggestion. We marked the figure and revised the description of this part in the article (Fig 4 has been changed to Fig 7).

4. Funding information is missing.

Thans for your question, this work was supported by Shandong Province Key R&D Program(Medical Food Special Program) Project(No.2019YYSP009), this information was included in the cover letter, as requested by plos one.

5. Declaration or conflict of interest is missing.

Thank you for your proposal. We added this part to the article.

---

## [Decision Letter · Decision Letter 1]

18 Nov 2024

Lyophilized powder of calf bone marrow hydrolysate liposomes improved renal anemia: In vitro  and in vivo  evaluation

PONE-D-24-31910R1

Dear Dr. Dai,

We’re pleased to inform you that your manuscript has been judged scientifically suitable for publication and will be formally accepted for publication once it meets all outstanding technical requirements.

Kind regards,

Salman Shakil

Academic Editor

PLOS ONE

Additional Editor Comments (optional):

Reviewers' comments:

Reviewer's Responses to Questions

**Comments to the Author**

1. If the authors have adequately addressed your comments raised in a previous round of review and you feel that this manuscript is now acceptable for publication, you may indicate that here to bypass the “Comments to the Author” section, enter your conflict of interest statement in the “Confidential to Editor” section, and submit your "Accept" recommendation.

Reviewer #1: All comments have been addressed

Reviewer #2: All comments have been addressed

2. Is the manuscript technically sound, and do the data support the conclusions?

Reviewer #1: Yes

Reviewer #2: Yes

3. Has the statistical analysis been performed appropriately and rigorously? 

Reviewer #1: (No Response)

Reviewer #2: Yes

4. Have the authors made all data underlying the findings in their manuscript fully available?

Reviewer #1: Yes

Reviewer #2: Yes

5. Is the manuscript presented in an intelligible fashion and written in standard English?

Reviewer #1: Yes

Reviewer #2: Yes

6. Review Comments to the Author

Reviewer #1: Thanks to the authors for upgrading the manuscript. The authors addressed all the comments and improved the manuscript as well. It can be accepted in the current form.

Reviewer #2: The authors address all the comments related to the major and minor concerns. They revised their manuscript according to the comments and now it will be publishable.

7. PLOS authors have the option to publish the peer review history of their article (what does this mean?). If published, this will include your full peer review and any attached files.

Reviewer #1: No

Reviewer #2: **Yes: **SHIHAB UDDIN

---

## [Editor Report · Acceptance letter]

22 Nov 2024

PONE-D-24-31910R1 

PLOS ONE

Dear Dr. Dai, 

I'm pleased to inform you that your manuscript has been deemed suitable for publication in PLOS ONE. Congratulations! Your manuscript is now being handed over to our production team.

Kind regards, 

on behalf of

Dr Salman Shakil 

Academic Editor

PLOS ONE